# The Low-Copy-Number Satellite DNAs of the Model Beetle *Tribolium castaneum*

**DOI:** 10.3390/genes14050999

**Published:** 2023-04-28

**Authors:** Tena Gržan, Mira Dombi, Evelin Despot-Slade, Damira Veseljak, Marin Volarić, Nevenka Meštrović, Miroslav Plohl, Brankica Mravinac

**Affiliations:** 1Ruđer Bošković Institute, Bijenička Cesta 54, HR-10000 Zagreb, Croatia; 2University Hospital Centre Zagreb, HR-10000 Zagreb, Croatia; 3Novo Nordisk Foundation Center for Protein Research, Faculty of Health and Medical Sciences, University of Copenhagen, DK-2200 Copenhagen, Denmark

**Keywords:** satellite DNA, tandem repeats, repetitive DNA, satellitome, *Tribolium castaneum*

## Abstract

The red flour beetle *Tribolium castaneum* is an important pest of stored agricultural products and the first beetle whose genome was sequenced. So far, one high-copy-number and ten moderate-copy-number satellite DNAs (satDNAs) have been described in the assembled part of its genome. In this work, we aimed to catalog the entire collection of *T. castaneum* satDNAs. We resequenced the genome using Illumina technology and predicted potential satDNAs via graph-based sequence clustering. In this way, we discovered 46 novel satDNAs that occupied a total of 2.1% of the genome and were, therefore, considered low-copy-number satellites. Their repeat units, preferentially 140–180 bp and 300–340 bp long, showed a high A + T composition ranging from 59.2 to 80.1%. In the current assembly, we annotated the majority of the low-copy-number satDNAs on one or a few chromosomes, discovering mainly transposable elements in their vicinity. The current assembly also revealed that many of the in silico predicted satDNAs were organized into short arrays not much longer than five consecutive repeats, and some of them also had numerous repeat units scattered throughout the genome. Although 20% of the unassembled genome sequence masked the genuine state, the predominance of scattered repeats for some low-copy satDNAs raises the question of whether these are essentially interspersed repeats that occur in tandem only sporadically, with the potential to be satDNA “seeds”.

## 1. Introduction

Eukaryotic genomes are rich in repetitive DNA sequences, among which transposable elements and satellite DNA (satDNA) sequences are the most abundant. Unlike transposable elements, which are scattered throughout the genome, repetitive units of satDNAs are arranged tandemly in arrays, sometimes encompassing megabase-sized regions. Although more than 60 years have passed since their discovery [1], satDNAs are still considered one of the greatest enigmas of the eukaryotic genome, especially with regard to their function. In many eukaryotes, they are located in regions of centromeric chromatin, suggesting their role in centromere determination and function [2]. Nevertheless, centromeric function cannot be their ubiquitous task since there are also satDNA-free centromeres [3]. It has also been shown that they can influence the activity of nearby genes [4]. In recent years, there has also been increasing evidence of changes in satDNA transcription associated with tumorigenesis [5,6]. Thus, different potential effects of satDNA have been observed in different organisms, but no satDNA function has yet been found to be universally present.

The discovery and experimental study of satDNAs have historically been hampered by their repetitive nature and predominant location in heterochromatin regions. In the pre-genomic era, restriction digestion of genomic DNA followed by gel electrophoresis was one way to discover the most abundant satDNAs in the genome of a species under study, which was also usually considered species-specific (reviewed in [7]). PCR facilitated the detection of less abundant satDNAs in a genome, as well as orthologous satDNAs among related species. With the advent of whole-genome sequencing and genome projects, the locations of these sequences outside heterochromatin regions are increasingly being discovered (for example, see [8,9]), although they have generally been largely omitted or underestimated in genome assemblies. High-throughput next-generation sequencing (NGS) of genomes resulted in a major step forward in the study of satDNAs, and the development of accompanying bioinformatics tools was crucial. One of the most frequently used computational pipelines for global analysis of repetitive DNAs from unassembled NGS reads without the need for genome assembly is the RepeatExplorer [10,11], which also integrates the TAREAN tool, specialized in the identification and characterization of satellite repeats [12]. RepeatExplorer and similar software have enabled in silico detection of numerous satDNAs in genomes. In most of the species analyzed in this way so far, several dozen satDNAs per genome have been discovered, but in some, significantly more have been found, such as 164 satDNAs in the fish *Megaleporinus macrocephalus* [13] and as many as 258 in the crayfish *Pontastacus leptodactylus* [14], as reviewed recently [15]. After revealing 62 satDNAs in the migratory locust [16], Ruiz-Ruano and co-authors proposed the term “satellitome”, which is now widely used by the community, to describe a whole set of different satDNAs in a genome. However, despite the increasing cataloging of numerous satDNAs, determining their location and large-scale arrangement in a genome is still a challenge. Applications of long-read sequencing methods, such as PacBio (Pacific Biosciences, Menlo Park, CA, USA) and ONT (Oxford Nanopore Technologies, Oxford, UK), are paving the way for assembling these tricky sequences [17], but the assembly of long satellite arrays is not yet routine.

In terms of evolution, related species may have the same or similar satDNAs. According to the concept of concerted evolution, when related species share a particular satDNA, monomeric repeats within a species show greater similarity than repeats between species [18]. The library hypothesis proposes the conservation of satDNA sequences, hypothesizing that related species share a common set of satDNAs differing in copy numbers but not in the DNA sequence [19,20]. Nowadays, with the increasingly detailed unraveling of satellitome collections in related species, evolutionary concepts and interpretations of the evolution of satDNA sequences are also being elaborated and extended [21].

The red flour beetle *T. castaneum* (Herbst, 1797) (Coleoptera: Tenebrionidae) is one of the most important cosmopolitan pests of stored grain products. Its potential for laboratory research was recognized five decades ago [22], and since then it has become a leading coleopteran model system for genetics and genomics studies associated with a wide range of fundamental and applied research (reviewed in [23]). As a representative of the Coleoptera, it was the first beetle to have its genome sequenced [24], and its most recent genome assembly, Tcas5.2, was recently improved with an updated genome annotation and a new official gene set [25]. However, although Tcas5.2 is a chromosome-level assembly, it lacks 20% of the genome sequence (44 Mb of 204 Mb). In addition, the karyotype of *T. castaneum* (2n = 20) counts 18 autosomal chromosomes and an Xy_p_ pair of sex chromosomes, but the genome sequence is anchored to 9 autosomes and the X, while the y_p_ chromosome remains unassembled. According to reassociation kinetics, nearly 40% of the *T. castaneum* genome consists of repetitive DNA [26], which is assumed to be the major component of the unassembled parts of the genome [27]. In *T. castaneum*, a major satDNA, TCAST, comprising 17% of the genome was discovered in the pre-genomic era with a restriction digestion approach [28]. TCAST is distributed in the (peri)centromeric heterochromatin of all 20 chromosomes [28], also extending over the regions of the functional centromere [29]. In addition to TCAST, ten other satDNAs were discovered 20 years later in the Sanger 7xdraft assembly Tcas3.0 [9,30]. However, the contributions of these satDNAs to the genome are much smaller and mostly do not exceed 1% for a single satDNA. Interestingly, unlike TCAST, these moderate-copy-number satellites are mainly present in euchromatic regions [9]. 

The aim of this work was to examine the entire satellitome of *T. castaneum* and, in particular, to determine the low-copy-number satDNAs that presumably exist in the genome but have not been revealed by the approaches used in previous studies. By employing low-coverage Illumina reads, we predicted potential satDNAs via assembly-free graph-based sequence clustering using the TAREAN tool. In addition to previously detected satDNAs, in this work, we detected 46 novel tandem repeats. By characterizing their distribution and organizational patterns using the current *T. castaneum* assembly, as well as unassembled scaffolds/singletons, we found 29 of them to be present exclusively in a tandem organization, while for 17 of them, the scattered repeats were also annotated. Experimental characterization of the three low-copy-number tandem repeats discovered in silico allowed us to validate them as satDNAs.

## 2. Materials and Methods

### 2.1. Insect Material and DNA Extraction

All experiments were performed using a laboratory culture of a highly inbred GA2 (Georgia 2) strain of the red flour beetle *T. castaneum* that was originally obtained from the USDA-ARS (Manhattan, KS, USA) in 2015. The insects were routinely reared in whole wheat flour at 28 °C and 50% relative humidity in the dark and subcultured every four weeks. Total DNA was extracted from 200 mg of adult beetles (100 individuals) using the DNeasy Blood and Tissue Kit (Qiagen, Hilden, Germany). DNA quantity was measured with a Qubit 4 fluorometer (Invitrogen, Waltham, MA, USA) using a Qubit dsDNA BR Assay Kit (Invitrogen, Waltham, MA, USA), while DNA quality was determined using a NanoDrop Spectrophotometer (Thermo Fisher Scientific, Waltham, MA, USA) and 1% agarose gel electrophoresis.

### 2.2. Illumina Sequencing and Graph-Based Clustering of Sequencing Reads

A *T. castaneum* genomic DNA library of ~500 bp fragments was prepared with a Kapa HyperPrep Kit (Roche, Basel, Switzerland) and the whole-genome sequencing (WGS) was performed on an Illumina NextSeq platform by Admera Health (South Plainfield, NJ, USA). The Illumina sequencing resulted in 8,122,518 paired-end reads (2 × 101 nt), and the raw sequencing data were deposited in the Sequence Read Archive (SRA) database under BioProject study accession number PRJNA606031. Quality control checks of raw Illumina reads were undertaken with FastQC [31]. FASTQ reads were preprocessed with the RepeatExplorer2 tools available at the Galaxy web-server (https://repeatexplorer-elixir.cerit-sc.cz/galaxy/ accessed on 25 August 2022). The reads were cleaned from adapter sequences, quality-filtered using quality scores ≥ 10 over 95% with no Ns allowed, and trimmed to 89 nt. The interlaced reads were randomly subsampled in order to scale down large datasets to low genome coverage. Seven randomly subsampled sets containing 230,000 to 2,300,000 reads corresponded to genome coverage of 0.1–1× (Appendix A). SatDNA identification via graph-based sequence clustering was performed using the TAREAN pipeline [12]. All TAREAN analyses were performed by applying the cluster merging option, a 0.0001 cluster size threshold, automatic filtering of abundant satellite repeats, DUST filtering, and an extra-long queue. Furthermore, in all TAREAN analyses, the custom repeat database containing previously identified *T. castaneum* satDNAs [9,28,30] was included to allow similarity comparisons of TAREAN-identified repeats with the previously described ones. In addition to automatic filtering of abundant satellite repeats, we also tested a set of 1,500,000 reads (0.65× genome coverage) from which we initially excluded reads that correspond to the highly abundant TCAST major satDNA [28]. TCAST sequences were removed from the reads using Geneious Prime v.2021.1.1. software, and eight TCAST sequences from NCBI (accession numbers U31088.1, U03758.1, U03753.1, U03752.1, KR046222.1, KR046221.1, KR046220.1, and HQ012048.1) were selected as references. Geneious Mapper was used to map to the reference with medium sensitivity and iterations of up to five times, resulting in 1,197,688 unused reads that represented sequences liberated of TCAST major satDNA and which were input for the TAREAN analysis T7 (Appendix A). Clusters containing satDNA were identified based on sphere- or ring-like-shaped graph topologies, and the consensus sequences of putative satDNA clusters (from both “high confidence” and “low confidence” ranks) from all TAREAN analyses were compared to each other using discontiguous Mega BLAST to disclose matching clusters of the same satDNA in different analyses. In Geneious Prime, the consensus sequences of putative satDNAs were annotated to the current *T. castaneum* reference genome assembly Tcas5.2 [25], deposited in the NCBI database under GenBank assembly accession number GCA_000002335.3. The Tcas5.2 assembly includes ten assembled linkage groups (GenBank accession numbers CM000276–CM000285) that correspond to ten chromosomes (LGX–LG10). We also analyzed the remaining sequences represented as unplaced scaffolds (GenBank accession numbers DS497665–DS497969) and unplaced singletons (GenBank accession numbers GG694051–GG695898), which derive from the unassembled part of the *T. castaneum* genome. A criterion for declaring a satDNA was an array of at least five consecutive copies with ≥70% similarity to a TAREAN cluster consensus present in the assembled or unassembled genome sequence. The consensus sequences of satDNAs detected in this work were used as the query sequences for BLAST searches against the NCBI nucleotide collection (https://blast.ncbi.nlm.nih.gov/Blast.cgi accessed on 14 September 2022) to check similarity with published sequences. Geneious Prime was used to perform local BLAST searches against the *Tribolium* RepeatScout library, which contains 4475 repeat families detected in *T. castaneum* categorized into several repetitive classes: HighA (31 families), HighB (5 families), Mid (304 families), Low (3237 families), 360 bp satellite (1), and transposable elements (897) [27]. The consensus sequences were also screened against Repbase, a database of prototypic sequences representing repetitive DNA from different eukaryotic species [32], which was accessed on 11 October 2022 (release 27.09) using CENSOR software [33]. The consensus sequences of the 46 novel satDNAs were deposited in the NCBI GenBank database under accession numbers OQ551416–OQ551461.

### 2.3. Computational Analyses of satDNAs

The satDNA monomer unit sequences annotated in the Tcas5.2 assembly and the unplaced scaffolds and singletons were extracted and used for further analyses. For satDNAs TCsat12, TCsat13, and TCsat14, monomeric unit sequences from the cloned fragments (see Section 2.4.) were also extracted and included in further analyses. For each satDNA, monomer sequences were aligned using Muscle v3.8.425 [34] with four iterations, anchor optimization, the kmer4_6 distance measure, UPGMB clustering, pseudo tree rooting, and the ClustalW weighting scheme. The obtained alignments were used to compute the basic nucleotide statistics, such as A + T content, monomer unit length, and pairwise identity. Relationships between satDNA abundance, monomer length, A + T content, nucleotide divergence, and monomer distribution were assessed using Spearman’s rank correlation coefficients. For satDNAs TCsat12, TCsat13, and TCsat14, the evolutionary relationships among monomer sequences were inferred by using the maximum likelihood (ML) method in MEGA11 [35]. The ML analyses were performed by using the nucleotide substitution models with the lowest Bayesian information criterion (BIC) scores, which were estimated to be the best: the Hasegawa–Kishino–Yano model with a discrete γ distribution for TCsat12, the Tamura three-parameter model with a discrete γ distribution for TCsat13, and the Tamura three-parameter model for TCsat14. The ML trees were reconstructed by applying the nearest-neighbor interchange (NNI) heuristic method, a very strong branch swap filter, and 1000 bootstrap replications, and the trees were visualized in FigTree v1.4.3 (https://github.com/rambaut/figtree/releases accessed on 19 October 2022). 

The genomic environment of low-copy satDNAs was examined for the presence of genes, transposable elements, and previously detected *T. castaneum* satDNAs [9,28,30]. First, satDNA locations were identified in the Tcas5.2 assembly [25] by using the BLAST algorithm and its interface in R, metablastr [36]. Only hits with >70% query coverage and >70% percentage identity were kept for further analysis. Repetitive and transposable elements were identified using the RepeatMasker program [37] on the Galaxy platform (usegalaxy.org) with the RepBase database and the “Hexapoda” species listing for clade-specific repeats. LINE, SINE, LTR, and similar transposable elements were filtered from the output to remove simple repeats, low-complexity regions, and other non-transposable elements. Gene locations were determined from the official gene set OGS3 for the Tcas5.2 assembly [25]. For each low-copy satDNA, its positions in the genome were first determined and then upstream and downstream flanking 5 kb regions were used for subsequent analysis. Quantification and visualization of the number of genes and transposable elements in the flanking regions were undertaken using custom scripts in R.

### 2.4. SatDNA Amplification, Cloning, and Sequencing

Based on the consensus sequences, specific primers for satDNAs TCsat12, TCsat13, and TCsat14 were designed using Primer3 software [38]. The fragments were amplified from *T. castaneum* genomic DNA by PCR in a 30 -µL reaction mixture containing 10 ng genomic DNA, 0.2 µM of each specific primer, 0.2 mM dNTP mix, 2.5 mM MgCl2, 0.25 U GoTaq G2 Flexi DNA polymerase, and 1× Colorless GoTaq Flexi Buffer (Promega, Madison, WI, USA). PCR amplification included a predenaturation step at 94 °C for 3 min, 35 amplification cycles (denaturation at 94 °C for 15 s, annealing at optimal temperature for 15 s, extension at 72 °C for 40 s), and a final extension at 72 °C for 10 min. The primers’ sequences, the optimal annealing temperatures, and the lengths of the amplified fragments for each satDNA are listed in Appendix A. PCR-amplified fragments corresponding to satDNA TCsat12, TCsat13, and TCsat14 dimers and trimers were cloned using the pGEM-T Easy Vector System (Promega, Madison, WI, USA) and XL10-Gold Ultracompetent Cells (Agilent Technologies, Santa Clara, CA, USA). Selection of potentially positive clones was undertaken using blue-white color screening for recombinant plasmids, and insert length verification was performed with colony-PCR amplification using the vector-specific primers M13F and M13R-40. Selected plasmid candidates were purified using a High Pure Plasmid Isolation Kit (Roche, Basel, Switzerland), and DNA inserts’ sequences were checked by sequencing (Macrogen Europe BV, Amsterdam, the Netherlands).

### 2.5. DNA Probes

DNA probes for Southern blot hybridization and fluorescence in situ hybridization (FISH) were generated from cloned satDNA fragments, the nucleotide sequences of which were verified by sequencing. In order to ensure the representation of different variants of the satellite monomeric sequence, a mixture of several different plasmid clones in an equimolar ratio was used in the preparation of each probe. Specifically, TCsat12-cl1/2/16/25/32 clones were used for the TCsat12 probe, TCsat13-cl27/32/40/47/50 clones for the TCsat13 probe, and TCsat14-2/10/31/32/34 clones for the TCsat14 probe (nucleotide sequences for specified clones are presented in Appendix A). DNA probes were labeled by PCR amplification using primers specific for each satDNA and the conditions mentioned in the previous section. The probes used in Southern blot hybridization were labeled with DIG-11-dUTP (Roche, Basel, Switzerland), while the probes used in FISH experiments were labeled with biotin-16-dUTP (Roche, Basel, Switzerland). After PCR labeling, probes were cleaned-up using a QIAquick PCR Purification Kit (Qiagen, Hilden, Germany).

### 2.6. Southern Blot Hybridization

For Southern blot analysis, genomic DNA was digested with different restriction endonucleases according to the manufacturer’s instructions (Thermo Fisher Scientific, Waltham, MA, USA). For each restriction digestion, 2 µg of genomic DNA was digested to completion with 20 U of restriction enzyme overnight at 37 °C. Digested genomic DNA was electrophoresed in a 1% agarose gel and blotted onto a positively charged nylon membrane (Roche, Basel, Switzerland). Hybridization was performed in the buffer containing 250 mM Na_2_HPO_4_ (pH 7.2), 20% SDS, 1 mM EDTA, 0.5% blocking reagent, and 20 ng/mL of the DIG-labeled specific probe with agitation overnight. Hybridization was carried out under high-stringency conditions at 68 °C, allowing >90% homology. Posthybridization washes were applied at 65 °C in the buffer containing 20 mM Na_2_HPO_4_, 1 mM EDTA, and 1% SDS. Chemiluminescent detection was undertaken using a DIG DNA Labeling and Detection Kit, CDP Star, and X-ray films (Roche, Basel, Switzerland).

### 2.7. Chromosome Preparations and Fluorescence In Situ Hybridization (FISH)

Localization of TCsat12, TCsat13, and TCsat14 satDNAs on the *T. castaneum* chromosomes was assessed using FISH. Chromosome spreads were prepared from pupae male gonads with the squash method as described previously [29]. FISH with biotin-labeled satDNA probes was performed for 18 h at 37 °C in a buffer containing 60% formamide, 2× SSC, 10% dextran sulfate, 20 mM Na_2_HPO_4_, and 10 ng/μL of the probe. Posthybridization washes were performed in 50% formamide/2× SSC at 37 °C. Biotin-labeled probes were detected with a fluorescein avidin D and biotinylated antiavidin D system (Vector Laboratories, Newark, CA, USA) via signal amplification through three layers of fluorophore conjugates and antibodies using the following dilutions: 1:500 fluorescein avidin D, 1:100 biotinylated anti-avidin D, and 1:2000 fluorescein avidin D. Finally, the slides were counterstained in 4′,6-diamidino-2-phenylindole (DAPI) solution for 15 min, air-dried, and embedded in Mowiol 4–88 mounting medium (Sigma-Aldrich, St- Louis, MO, USA). Slide visualization and image acquisition were undertaken using a Leica TCS SP8 X confocal laser scanning microscope (Leica Microsystems, Wetzlar, Germany) equipped with an HC PL APO CS2 63×/1.40 oil objective, 405 nm diode laser, and a supercontinuum excitation laser (Leica Microsystems, Wetzlar, Germany). Images were captured separately for each fluorochrome and processed with ImageJ [39] and Adobe Photoshop CS5 (Adobe Systems, San Jose, CA, USA).

## 3. Results

### 3.1. High-Throughput Search for satDNAs

To thoroughly explore the content of the *T. castaneum* satellitome, we sequenced the whole genome of the flour beetle *T. castaneum* de novo using Illumina sequencing. Randomly subsampled sets of Illumina reads were used in seven series of TAREAN clustering (Appendix A). For the first six analyses (T1–T6), we used the sets of input reads that statistically provided genome coverage of 0.1×, 0.25×, 0.5×, 0.65×, 0.75×, and 1× (Appendix A). As expected, depending on the number of input reads used, the analyses resulted in different numbers of clusters being reported as putative satDNA repeats ranging from 12 to 72, with the maximum being reached when analyzing 2,300,000 reads, corresponding to 1× genome coverage (T6 analysis, Appendix A). However, we found that the number of clusters classified as highly putative satDNAs reached its plateau in the analysis of 1,500,000 reads (T4 analysis, Appendix A), so we concluded that 0.65× genome coverage might represent the optimal trade-off between the maximum number of reads used and possible oversaturation of the analysis. It should be emphasized that all TAREAN analyses were executed with the “perform automatic filtering of abundant satellite repeats” option, and they were highly consistent in estimating the proportion of the major satDNA TCAST in the genome to be 17–18%, as estimated experimentally [28]. Since it is known that highly abundant sequences can affect the sensitivity of an analysis in terms of inefficient detection of low-copy satDNAs, we wanted to check whether the clustering would be significantly different if we used the initial dataset without the TCAST satellite. For this reason, we first filtered out TCAST from a set of 1,500,000 reads (0.65× genome coverage) and then performed a TAREAN analysis using the remaining 1,197,688 TCAST-free reads (T7 analysis, Appendix A). Although clustering of the TCAST-free dataset did not predict more satDNAs in general, it resulted in a higher number of highly putative satDNAs (Appendix A). Since T4 and T7 analyses using 0.65× genome coverage were deemed most suitable for predicting potential satDNAs, further analyses were performed using their clusters.

The consensus sequences of clusters annotated as putative satDNAs in T4 and T7 analyses were BLAST-searched against the NCBI GenBank nucleotide database. In this way, we identified the clusters corresponding to the 11 previously described satDNAs [9,28,30] and excluded them from further analysis. We also excluded the cluster representing *T. castaneum* mitochondrial DNA, which matched GenBank entry KM009121 with 99.73% identity (E-value 0.0) over the entire length of the sequence (15,884 bp). Using the sequence similarity criterion of ≥70%, we annotated the consensus sequences of the remaining clusters in the current *T. castaneum* genome assembly Tcas5.2, which consists of ten chromosome/linkage groups [25]. We also annotated the consensus sequences in the unassembled portion of the genome, which includes 305 unplaced scaffolds and 1848 unplaced singletons. When a particular sequence was tandemly repeated in an array of ≥5 monomers, either in the assembled or unassembled part of the genome, the cluster was declared as a potential satDNA. In this way, we identified 46 novel satDNAs (Table 1, Appendix A). The average number of tandem repeats in the annotated arrays was 18, whereas the numbers of repeats in the longest arrays for individual satDNAs ranged from 5 to 84 (Appendix A). In addition to arrays of ≥5 consecutive monomers, we also annotated scattered copies in the form of short segments consisting of <5 repeat units for 17 satDNAs (Table 1 and Appendix A).

Among the 46 novel satDNAs, 9 showed partial matches to repeat elements recorded in the *Tribolium* RepeatScout library, sharing similarity with 1 HighB, 5 Mid, and 3 Low repetitive elements, annotated according to [27] (Appendix A). A search for identity against repeats of the Repbase collection did not yield any complete sequence hits with deposited repetitive DNA elements, although 26 satDNAs showed partial similarities with segments of 37 transposable elements (Appendix A), among which DNA transposons predominated (Appendix A). The best alignment score was obtained for one satDNA with a 164 bp long repeat unit present in the inverted termini of the 9.5 kb long Rehavkus -1_TC DNA transposon (Appendix A).

### 3.2. Characteristics of the Low-Copy satDNAs

The novel satDNAs were named TCsats, and satellite catalog numbers were assigned based on the cluster incidence in the TAREAN analyses and the genomic abundance. Ruiz-Ruano and co-authors [16] have suggested that, when additional satDNA families are found in a species, they should be numbered sequentially from the last one described in previous work. Following this recommendation, we cataloged newly discovered satDNAs from TCsat12 to TCsat57 (Table 1). In Table 1, the 11 previously characterized satDNAs for *T. castaneum* [9,28,30] are listed as TCsat1–TCsat11, simply for the purpose of sequential numeration of novel satDNAs, but we retain their original names in further references.

The 46 newly identified satDNAs made up 80.7% of the total number of satDNAs identified in *T. castaneum*; nevertheless, they comprise only 2.1% of the *T. castaneum* genome. Moreover, satDNA TCsat15 alone accounts for 1.7% (Table 1), reducing the contribution of the remaining 45 satDNAs to only 0.4% of the genome. Considering that 45 of the 46 newly discovered satellites are individually represented by less than 0.1% in the genome (on average, 0.01% per satDNA), they can indeed be considered low-copy-number satDNAs in the satDNA-rich *T. castaneum* genome.

The satDNAs identified in this work showed a wide range of monomer lengths from 73 bp to 1106 bp (Table 1 and Appendix A). However, two length ranges were favored, 140–180 bp and 300–340 bp, with the first range being more preferred (Figure 1A). The inclination towards 140–180 bp long repeat units is consistent with the preferences of the *T. castaneum* satDNAs described previously (Figure 1A) [9,28,30]. Regarding base composition, in the pool of newly described satellites, all but one had an A + T content > 60% and more than half of them (27 of 46) showed a tendency towards an A + T composition ≥ 70% (Figure 1B).

The monomers of the 46 satDNAs annotated in the Tcas5.2 assembly and unplaced scaffolds/singletons were extracted (Appendix A) and aligned. Based on the alignments, the 46 satDNAs showed a wide range of sequence divergence, ranging from 1.3% to 33.9% (Appendix A). Nevertheless, according to Spearman’s rank statistics, divergence showed no correlation with satDNA abundance (rs = 0.242, *p* = 0.10), monomer length (rs = −0.096, *p* = 0.52), or A + T content (rs = −0.165, *p* = 0.27).

Analyzing the highly variable satDNAs, we found that some of them consisted of different subfamilies or had a complex monomeric structure. Among the newly discovered satDNAs, TCsat37 showed the highest nucleotide divergence of 33.9%. Its highly diverse monomers fell into three subfamilies distinctive in their average monomer lengths: 211 bp, 270 bp, and 330 bp. While the divergence between consensus sequences of these subfamilies ranged from 29.4% to 35.8% (Appendix A), the divergence of the monomers within each subfamily was much lower, ranging from 2.6% to 4.8%. Although the monomers of all three subfamilies were scattered across a number of chromosomes, arrays of tandemly repeated monomers of a particular subfamily were observed at some locations. For example, TCsat37–211 monomers were homogenized on the unplaced singletons GG694714 and GG694827, TCsat37–270 monomers on chromosome chLG5, and TCsat37–330 monomers on chromosome chLG7 (Appendix A). This observation suggests that some arrays may have been propagated recently or are more likely to be influenced by mechanisms that favor their homogenization. Another highly variable satDNA was TCsat38, with a nucleotide divergence of 26.1%. In addition to the variable scattered monomers, the divergence of TCsat38 was due to the fact that its tandem arrays are built on dimeric higher-order repeats (HORs), the subunits A and B of which differ by 19.4% on average (Appendix A), whereas the variability among monomers of the same subunit group was two times lower (10.5% and 8.7% for subunit groups A and B, respectively). The satDNA TCsat18, with 25.4% divergence and based on ~73 bp monomers, was the *T. castaneum* satDNA with the shortest repeat units (Table 1). Although 73 bp long monomers dominated among the TCsat18 repeats, two other monomer variants 63 bp and 82 bp long were identified (Appendix A). The 73 bp variant and the 63 bp variant shared 82.25% similarity, and the more divergent 83 bp variant showed duplication of a monomer segment (Appendix A). However, in contrast to the 73 bp variant, the 63 bp variant and the 83 bp variant did not form homogenous tandem arrays but occurred in combination with the 73 bp variant, forming dimeric HOR units. Thus, 73 bp + 82 bp HORs were detected on chromosome chLG4, whereas 73 bp + 63 bp HORs predominated on chromosome chLG9 (Appendix A). The satDNA TCsat15 was another highly variable satellite that not only exhibited high divergence (19.5%) but also stood out from other low-copy satDNAs due to its genome abundance (1.7%) and repeat unit length of 1106 bp (Table 1). TCsat15 was also the only novel satellite present on all chromosomes (Table 1), mostly in the form of scattered divergent repeat units present in arrays of <5 consecutive monomers. In addition, chromosomes chLG3, chLG7, and chLG8 harbored 1277 bp long repeat units composed of 1106 bp long TCsat15 monomers and a 171 bp segment that matched with ~80% similarity the sequence of the previously described satDNA Cast2 [9] (Appendix A). The combination of the TCsat15 monomer and Cast2 sequence was also identified in many scattered copies, which may indicate a form of coevolution of these two repetitive sequences. In addition, the TCsat15 monomers contained a 164 bp long segment that showed ~65% similarity to TFREE (Appendix A), the major satDNA of the sibling species *Tribolium freemani* [40]. The partial similarity to the satDNA of the congeneric species might imply the possibility of the complex evolution of TCsat15, the satDNA that appears to differ in many aspects from other *T. castaneum* satellites.

### 3.3. Organization of the Low-Copy satDNAs

Mapping of the 46 novel satDNAs to the current reference genome Tcas5.2 disclosed that the monomers of 29 satDNAs were exclusively on the ten assembled chromosomes and four satDNAs were mapped only to the unplaced scaffolds and singletons, while 13 satDNAs were detected in both the assembled and unassembled genome sequences (Table 1 and Figure 2A). The largest number of satDNAs, 21, was identified on the largest chromosome chLG3 of *T. castaneum* (Figure 2B), but no correlation was observed between the size of the other chromosomes and the number of different satDNA families detected on them. In the current assembly, the majority of low-copy satDNAs (31 of them) were annotated on one chromosome only or on unplaced scaffolds/singletons (Table 1). In this feature, the low-copy satDNAs differ from the previously described satDNAs [9,28,30], which are distributed across six to ten chromosomes (Table 1). However, it must be emphasized that 17 low-copy satDNAs were identified in unplaced scaffolds and singletons (Figure 2B) with positions in the genome that have not yet been determined since 20% of the assembled genome sequence is missing [24,25].

Despite the incompleteness of the current genome assembly, we tried to decipher the organization of low-copy satellites based on available data and annotated monomer copies. First, we wanted to determine the extent to which the low-copy satDNA monomers were tandemly repeated or scattered as individual copies. Using ≥5 consecutively repeated monomers as a criterion for tandem organization, we found 29 satellites that had monomers exclusively repeated in tandem (Table 1 and Appendix A), whereas, for 17 satDNAs, the scattered monomers or short stretches of <5 contiguous repeats were also detected. In 13 of these 17 satDNAs, repeats found as single monomers or as a few tandem repeats (<5) accounted for less than 50% of the detected copies (Appendix A). Only in four satDNAs (TCsat15, TCsat18, TCsat24, and TCsat37) did the proportion of scattered annotated copies exceed 50%. However, these were satDNAs for which significantly fewer copies were annotated in the assembled genome than would be expected from the TAREAN estimation of genomic abundance (Appendix A), and there is therefore the possibility that greater numbers of their tandem copies remained hidden in the unassembled part of the genome. Further, it was noticeable that the satellites TCsat15, TCsat18, TCsat37, and TCsat38, which were distributed over the largest numbers of chromosomes (Table 1 and Appendix A), also had high proportions of scattered monomers (Appendix A). When all 46 satDNAs were considered, a strong positive correlation (rs = 0.684, *p* < 0.001) was found between the proportion of scattered repeats and the distribution across different chromosomes and unassembled sequences. The proportion of scattered repeats also showed a positive correlation with satDNA abundance (rs = 0.529, *p* < 0.001), as well as with nucleotide divergence (rs = 0.559, *p* < 0.001) (Appendix A). No correlation, however, was found between the dispersion of monomers and the length of monomer units (rs = 0.104, *p* = 0.494).

Next, we aimed to gain insight into the genomic environment of low-copy satDNAs. The analysis focused on the assembled part of the Tcas5.2 genome because it has well-curated annotations. We searched for the presence of genes, transposable elements, and previously described satDNAs [9,28,30] in the vicinity of the newly discovered satDNAs. As mentioned previously, 4 of the 46 novel satDNAs (TCsat13, TCsat16, TCsat31, and TCsat54) were not detected in the assembled genome Tcas5.2. For the 42 novel satDNAs that were detected in the assembly, we examined the 5 kb flanking regions of their tandemly organized and their scattered repeats. TCsat44 was the only one that did not have any of the three addressed sequence types in the surrounding 5 kb regions; therefore, it is not presented in Figure 3. For the remaining 41 satDNAs, satellite repeats were detected in the flanking regions of 16 satDNAs, 35 satDNAs had annotated genes in their vicinity, and all 41 satDNAs had transposable elements in the 5 kb surrounding regions (Figure 3). In other words, there were no low-copy satDNAs that exclusively had satellite repeats or genes or a combination of these two sequence types in their proximal regions. On the other hand, there were six satDNAs with 5 kb surrounding regions in which only transposable elements were found. Among the transposable elements, DNA transposons were detected most frequently (Appendix A). They were found ten times more frequently than long interspersed nuclear elements (LINEs), which are the most frequently found among retrotransposons (Appendix A). Regarding the genes detected in the flanking regions of 35 satDNAs, 92.5% of them were associated with hypothetical and uncurated proteins according to the OGS3 annotation (Appendix A). Among the 43 genes that had a defined function, we did not detect any grouping with respect to their function. Regarding the previously described satellites discovered in the vicinity of the 16 novel satDNAs, 90.2% of the satellite annotations were located near the scattered copies of the satDNA TCsat15 (Appendix A). In the majority of these cases (91.1%), Cast2 was detected near TCsat15, in accordance with the previously described association of these two satDNAs. As for the other satellite DNAs, they were detected in a much smaller number of cases (Appendix A).

### 3.4. Experimental Analysis of the Three Low-Copy satDNAs

In addition to the in silico analysis of the *T. castaneum* satellitome, we experimentally characterized three novel satDNAs with clusters that were declared as putative satellites in all TAREAN analyses: TCsat12, TCsat13, and TCsat14 (Appendix A). 

First, we created specific primers for each satDNA and amplified the fragments from genomic DNA using PCR (Appendix A). Electrophoresis of the PCR products for all three satellites revealed a ladder-like profile characteristic of tandemly repeated sequences (Appendix A). The PCR-amplified dimers and trimers were cloned and sequenced, and the sequences of 31–46 monomers were extracted from the sequenced dimers/trimers (Appendix A). The extracted monomers obtained by PCR were aligned with the monomers annotated in the genome assembly and the unplaced scaffolds/singletons (Appendix A). The phylogenetic analyses for all three satDNAs did not show any preferential grouping of monomers when considering their origin in terms of genome outset or PCR provenience. The presence of PCR-derived monomers that intermingled with genome-annotated monomers in the phylogenetic trees (Figure 4) proved that the PCR amplification was not biased and did not propagate a subset of monomer variants, as well as confirming that the TAREAN consensus sequences from which the PCR primers were generated were truly representative. Nevertheless, the phylogenetic analyses revealed which monomers were most divergent. Interestingly, in the case of TCsat12, which had tandem arrays in unplaced scaffolds/singletons and only four scattered copies on chromosome chLG6, these four copies from chromosome chLG6 formed a divergent, separate group supported by a 95% bootstrap value (Figure 4A). Similarly, in the case of TCsat14, which has tandem arrays on chLG7 and only one copy on chromosome chLG3 plus one on an unplaced scaffold, these two remote single copies clearly departed from the rest of the tandemly organized monomers (100% bootstrap value) (Figure 4C). The increased divergence and remote locations of these copies implied that they may be distant, isolated copies and could possibly be subject to independent sequence evolution. 

Based on the consensus sequences TCsat12, TCsat13, and TCsat14, we selected restriction endonucleases (REs) that had only one cutting site in a monomeric sequence (Appendix A). Digestion of genomic DNA with the selected REs followed by Southern blot hybridization with specific satDNA probes resulted in a ladder-like hybridization pattern for all three satDNAs, corresponding to the arrangement of monomeric fragments and their multimers (Figure 5). While we found only one commercially available RE that cut once in the TCsat12 consensus sequence (Figure 5A), the range of REs for TCsat13 and TCsat14 was larger (Figure 5B,C). Digestion of TCsat13 arrays with four different REs yielded the expected 266 bp-based ladder-like pattern but with some differences between digestions resulting from additional or mutated RE sites in some monomers (Figure 5B). For example, the absence of the *Hae*III restriction site in 54.8% of the analyzed TCsat13 monomers (Appendix A) explained the weak monomeric signal in the *Hae*III digest reaction and the “higher ladder” profile (Figure 5B). On the other hand, TCsat14 was the most homogeneous *T. castaneum* satDNA, showing only 1.3% divergence (Appendix A). Therefore, it is not surprising that digestions with the REs *Alu*I, *Rsa*I, and *Pvu*II, the target sites of which were conserved in 97.6–100% of the analyzed monomers (Appendix A), revealed a hybridization signal that was almost exclusively related to monomeric fragments (Figure 5C). However, when we digested TCsat14 arrays with *Hin*cII, the restriction site of which was detected in only 54.2% of monomers (Appendix A), the hybridization signals showed a ladder-like profile (Figure 5C). In summary, Southern blot hybridization confirmed the tandem organization of the satDNAs studied, and the hybridization profiles obtained in different digestion reactions were consistent with the mutations diagnosed with in silico restriction-site analyses.

When we mapped TCsat12, TCsat13, and TCsat14 to the current assembly, TCsat14 tandem arrays were annotated on the chromosome chLG7 sequence, but TCsat12 and TCsat13 tandem arrays were detected only in unplaced scaffolds and singletons (Table 1). To determine their positions on *T. castaneum* chromosomes, we performed FISH using the cloned PCR fragments as specific probes. On the haploid complement, FISH mapping of TCsat12 revealed its presence on the two chromosomes, with a consistently stronger signal on one of them (Figure 6A). The very similar sizes of the six *T. castaneum* autosomes prevented their accurate differentiation [41]; nevertheless, based on the examined spreads, it can be concluded that TCsat12 arrays were not located on the three largest autosomes chLG2, chLG3, and chLG4 (Figure 6A). FISH analysis with the TCsat13-specific probe showed the presence of the TCsat13 signal on one chromosome in the haploid set (Figure 6B). One signal was also detected upon hybridization with the TCsat14-specific probe (Figure 6C), and according to the in silico annotation, this signal was located on chromosome chLG7 (Appendix A). On the spreads of less condensed chromosomes, we found that the TCsat14 signal was located at the chromosome tip and consisted of two very close dots that merged into one signal on most of the observed spreads (Appendix A). The presence of two closely spaced signals was consistent with the assembled genome, according to which two TCsat14 satDNA arrays (3.4 kb and 7.4 kb long) separated by 65 kb are placed at the beginning of chromosome chLG7 (Appendix A). For all three satDNAs, it is very likely that the FISH analyses revealed the positions of the most prominent tandem arrays, and potentially shorter arrays consisting of fewer or scattered monomers were not visualized because they were below the FISH detection limits.

## 4. Discussion

The genome of the representative coleopteran species *T. castaneum* is prominently marked by a high abundance of the (peri)centromeric satDNA TCAST, which was discovered nearly three decades ago [28]. This finding postdates discoveries of satDNAs in the congeneric species *Tribolium confusum* [42] and *T. freemani* [40], in which abundant but unrelated species-specific (peri)centromeric satDNAs were detected making up 40% and 30% of the respective genomes. The characterizations of satDNAs in five other *Tribolium* species that followed in subsequent years and were also based on restriction digestions of genomic DNA demonstrated that one or two highly copious and species-specific satDNAs dominate in other *Tribolium* genomes as well [43,44,45,46]. However, with the availability of the *T. castaneum* genome assembly, it has become clear that TCAST is not the only satDNA in *T. castaneum*, as ten other satDNAs 20–85 times less abundant than TCAST have been found [9,30]. The progress in sequencing and computational pipelines has recently led to the discovery of numerous satDNAs in insect genomes. For example, about 30 satDNAs have been detected in ladybird beetles [47] and kissing bugs [48], more than 60 satellites in grasshopper genomes [16,49], and 112 satellites in the red palm weevil [50], while as many as 160 satellites were identified in the triatomine bug *Triatoma delpontei* [51]. Therefore, it was logical to ask whether there are more satDNAs in *T. castaneum* than the 11 previously characterized, and cataloging its satellitome became the main objective of this research.

To address this task, we applied Illumina sequencing and took advantage of satDNA detection without the need for genome assembly. In this work, a quarter century after the discovery of the first *T. castaneum* satDNA, we identified 46 new satDNAs in the genome of this important model insect. This expands the *T. castaneum* satellitome to a total of 57 satDNAs, which cumulatively make up 23.8% of the genomic sequence. The most striking difference between the newly discovered satDNAs and those previously characterized is their low proportions in the genome, with 45 satellites each accounting for less than 0.1% and thus being designated as low-copy-number satDNAs. The *T. castaneum* satellitome, in which 79% of the members are low-copy satDNAs, is in line with many insect satellitomes, in which 70–80% of the total numbers of detected satellites belong to satDNAs that individually make up less than 0.1% of the genomes [47,48,52].

Despite the large difference in abundance, the newly described satDNAs share some common features with already known *T. castaneum* satellites. One of them is the high A + T content of the repeat units, which predominantly ranges between 60 and 80%. High A + T composition is a frequent characteristic of satDNAs in many organisms, and it has been suggested that AA/TT periodicity in satellite monomers may contribute to nucleosome stabilization (reviewed in [53]). However, the *T. castaneum* genome was found to be A + T-rich (67% according to [24], as also confirmed by our Illumina WGS data), so it is possible that *T. castanuem* A + T-rich satDNAs simply reflect the overall A + T richness of the genome. Another feature in which the newly discovered satDNAs are similar to those described previously is the tendency for the repeat unit length to be 140–180 bp or 300–340 bp. This property, recognized as the DNA length required to wrap around one or two nucleosomes, has been observed in many centromeric satDNAs [54,55]. Interestingly, in *T. castaneum*, the aforementioned repeat unit lengths are favored not only by the highly abundant, centromeric TCAST satellite located in heterochromatic (peri)centromeric regions [28,29] but also by moderate-copy satDNAs distributed primarily along euchromatic chromosomal arms [9], as well as by most low-copy satDNAs. In other words, the *T. castaneum* satellitome contains satDNAs, which, despite drastic differences in abundance and different chromosomal positions and chromatin environments, incline to high A + T compositions and preferred lengths for repeat units. It could be hypothesized that the low-copy satellites included in the *T. castaneum* satellitome, having certain structural sequence attributes, might have the potential to be propagated into more abundant satDNAs under certain evolutionary circumstances. 

The feature that clearly distinguishes the low-copy satDNAs discovered in this work from the previously known *T. castaneum* satellites is the chromosomal distribution. In contrast to the previously described satDNAs, the copies of which are annotated on most chromosomes, in the current Tcas5.2 assembly, we mainly detected most low-copy satDNAs on one or only a few chromosomes. It is intuitive to expect that low-represented satDNAs will not be distributed over a larger number of chromosomes, and the presence of low-copy satDNA families restricted to one or only a few chromosomes has also been evidenced in other organisms [16,49,56]. In the *T. castaneum* genome, which abounds in large heterochromatin blocks, the low-copy satellites are nevertheless mainly embedded in euchromatic regions, as was also found for its moderate-copy satDNAs [9]. Recent satellitome studies of the Coleoptera species *Hippodamia variegata* [47] and *Rhynchophorus ferrugineus* [50] have shown that, as in the case of *T. castaneum*, their genomes are dominated by a major, highly abundant satDNA that is primarily, although not exclusively, distributed in the (peri)centromeric heterochromatin of all chromosomes, while low-copy satDNAs mostly have a scattered and sporadic distribution along euchromatic regions. That low-copy satDNAs can successfully exist outside of heterochromatic regions has also been confirmed in heterochromatin-poor organisms, such as the Pacific oyster *Crassostrea gigas*, in which short satDNA arrays, often incorporated into Helitron/Helentron mobile elements, are widely dispersed throughout the genome [57,58]. One might hypothesize that some of the low-copy satDNAs arrays scattered along euchromatic regions represent “satDNA seeds” that have the potential to be amplified into more extensive satDNA families and that euchromatin, being less inert than heterochromatin, may even facilitate their propagation. In *Drosophila virilis* and *Drosophila americana*, the transition zones between densely condensed heterochromatin and lightly packed euchromatin were detected as regions of recently formed, abundant tandem repeats derived from DNA transposons, suggesting that the less compacted environment of the higher recombination rates may favor amplification of satDNA arrays [59,60]. 

Transposons dominated in the analyzed genomic environment of *T. castaneum* low-copy satDNAs. Transposable elements are also a type of sequence with which certain *T. castaneum* low-copy satDNAs share similarities in some segments of their repeat units. The link between satDNAs and transposable elements has been extensively demonstrated and elaborated in the literature [15,61,62]. There are numerous indications that different classes of transposable elements serve as sources for the formation of novel satDNAs. In *D. virilis*, for example, tandem repeats within the Tetris DNA transposon led to the TIR-220 satDNA [59]. A recent study of the satDNA-rich legume *Lathyrus sativus* showed that 9 of the 11 most abundant satDNAs also had short tandem arrays located within LTR-retrotransposons [63]. For *T. castaneum*, TCAST-like sequences were found to be embedded in a complex unit resembling a DNA transposon, but TCAST-like elements were also detected within the CR1-3_TCa non-LTR retrotransposon [64]. In this work, we found that the low-copy satellite TCsat23 occurred in the Rehavkus -1_TC transposon. Several ~95% identical copies of this DNA transposon have been detected in the *T. castaneum* genome [65], and TCsat23 forms short tandem arrays of mostly five and eight consecutive monomers in its inverted termini. SatDNA repeats, which also occur as short segments in transposable elements, raise the question of whether the evolutionary forerunners are long tandem arrays or transposon elements carrying a smaller number of their repeats [63]. Scalvenzi and Pollet [66] suggested a model according to which transposition of mobile elements carrying a satellite motif could lead to the amplification and spread of tandem repeats and become driven by increased recombination rates as the number of tandem arrays within the mobile element expands. Eventually, tandem repeats originating from mobile elements bring forth classical, standalone satDNA arrays. For TCsat23, we found stretches no longer than nine consecutive monomers in the current assembly, so we assumed that this satDNA was transposon-derived. Of course, we should not discount the fact that a significant part of the *T. castaneum* genome is still missing, and perhaps ultra-long sequencing will reveal the genome regions that contain longer TCsat23 arrays. Long-read sequencing, which unravels extensive tandem arrays, and comparison of related genomes are promising tools for deciphering the origin of satellites. Applying such an approach to the plant *L. sativus*, it was deduced that its three satDNAs evolved from the tandem subrepeats originally contained in the LTR retrotransposon sequence [63].

Within the frame of *T. castaneum* satellitome characterization, we experimentally evaluated three low-copy satDNAs predicted by TAREAN. Despite their low representation in the genome (0.015–0.046%), Southern hybridization successfully confirmed their tandem organization and yielded hybridization profiles consistent with the variability calculated from PCR-amplified fragments and monomers annotated in the genome assembly. In addition to the 46 satDNAs cataloged in this work, TAREAN also proposed a certain number of clusters as potential satellites for which we could not annotate tandemly organized copies either in the current genome assembly or among the unplaced scaffolds/singletons, so we did not include them in the satellitome collection. We assume that if these potential sequences have tandemly organized repeats, they are placed in the 20% of the genome sequence that has not yet been assembled. It is expected that future long-read sequencing will disclose whether these are indeed satDNAs, and it is possible that the current satellitome collection will be expanded. The long-read sequencing will also reveal whether some of the low-copy-number satDNAs discovered in this work form longer tandem arrays than those we were able to annotate in the current assembly.

To define the *T. castaneum* satellitome, we resequenced the genomic DNA of the highly inbred GA2 strain used in the *T. castaneum* genome sequencing project [24] and on which the Tcas5.2 reference assembly is based [25]. It should be kept in mind, however, that there may be differences in satDNA profiles between individuals and also between populations within a species. For example, the two main evolutionary lineages of the hemipteran insect *Triatoma infestans*, although they share most of the satDNAs detected in their genomes, show significant differences in the representativeness of certain satDNA families [51,67]. In *T. castaneum*, the population-specific profiles were found for the three subfamilies of the satDNA TCAST2 [30]. The *T. castaneum* collection of 57 satDNAs presented here is an initial reference, but it is possible that future analyses of other strains/populations will unveil the plasticity of the *T. castaneum* satellitome in terms of the number of satDNA families and/or their genomic abundance.

The advent of ultra-long-sequencing technologies and the complete T2T-CHM13 human genome sequence have ushered in the telomere-to-telomere (T2T) genomics era [68]. Although repetitive DNA regions remain the last genome bastions that most strongly resist correct assembly, better sequencing techniques, as well as improvements in bioinformatics assembly tools, are bringing us closer to the moment when we will also be able to achieve gapless, high-quality T2T assembled genomes for other species. The availability of T2T genomes will enable comprehensive compiling of satDNAs and detailed insights into their long-range organization and sequence variability. From this information, we will be able to learn more about the origin, evolution, and functional roles of these unusual sequences. Once its highly repetitive genome is fully assembled, the model beetle *T. castaneum* will certainly help to approach these findings.

## Figures and Tables

**Figure 1 genes-14-00999-f001:**
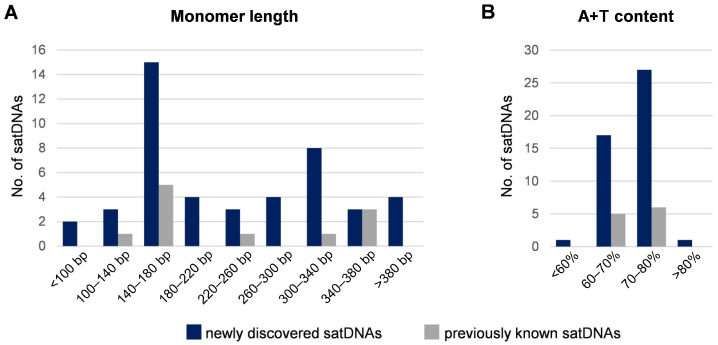
Monomer length distribution (**A**) and A + T content (**B**) of *T. castaneum* satDNA consensus sequences. The dark blue bars represent the 46 satDNAs identified in this work, while 11 previously described satDNAs are indicated by gray bars.

**Figure 2 genes-14-00999-f002:**
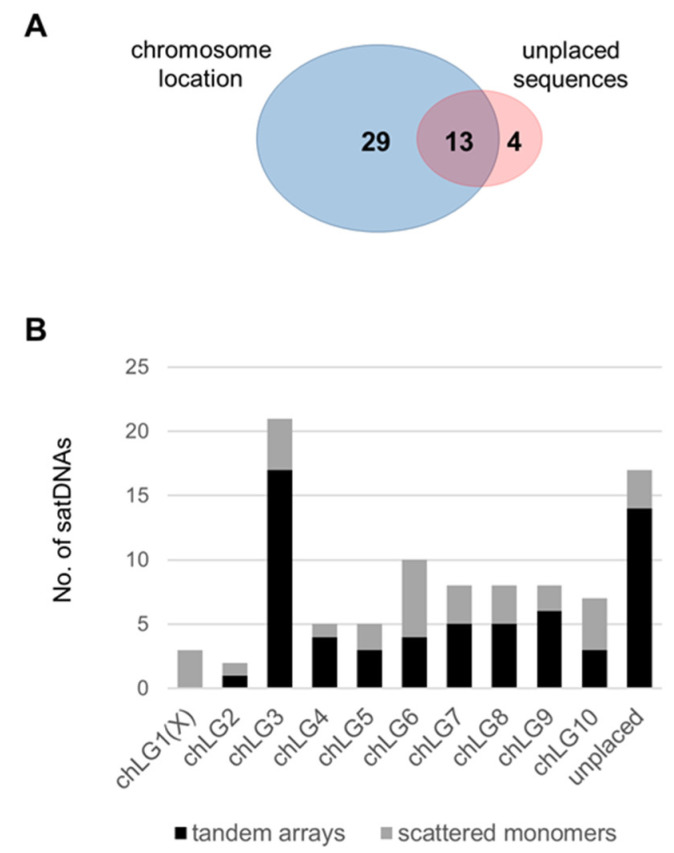
Distribution of the 46 novel satDNAs annotated in the *T. castaneum* genome assembly Tcas5.2. (**A**) Venn diagram showing the number of satDNAs annotated at the assembled chromosomes (blue) and the unplaced scaffolds/contigs (red). (**B**) The numbers of satDNAs annotated on the ten chromosomes chLG1–chLG10 and the unplaced scaffolds/contigs present in arrays of ≥5 tandem monomers (black bars) or in the form of scattered monomers (gray bars).

**Figure 3 genes-14-00999-f003:**
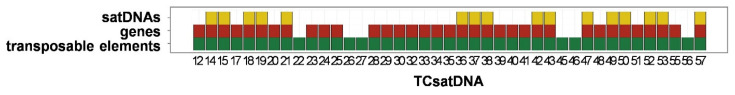
The presence of previously described satDNAs [9,28,30], genes, and transposable elements in the 5 kb flanking regions of the low-copy satDNAs detected in this work. The presence of satDNAs, genes, and transposable elements is indicated by yellow, red, and green squares, respectively. The lists of identified genes, transposable elements, and satDNAs are presented in Appendix A.

**Figure 4 genes-14-00999-f004:**
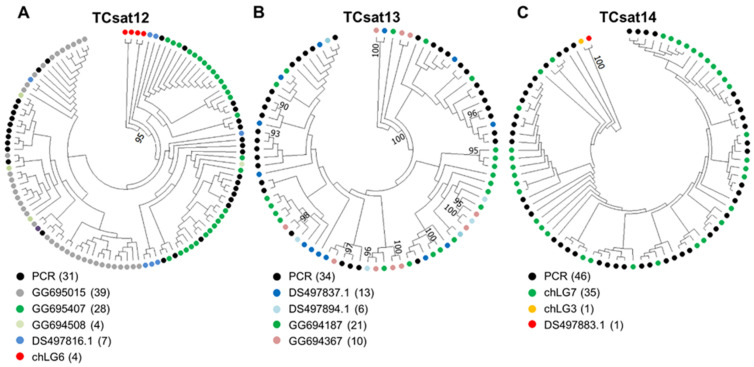
Maximum-likelihood trees showing the relationships between monomer sequences TCsat12 (**A**), TCsat13 (**B**), and TCsat14 (**C**). PCR-amplified copies are marked by black dots, whereas monomer copies annotated at chromosomes (chLG) and unplaced scaffolds and singletons (DS and GG, respectively) are marked by color-coded dots. The number of copies from a given source is indicated in parentheses in the color-code legend. The nodal supports were calculated based on 1000 bootstrap replicates, and only bootstrap values ≥ 90 are shown.

**Figure 5 genes-14-00999-f005:**
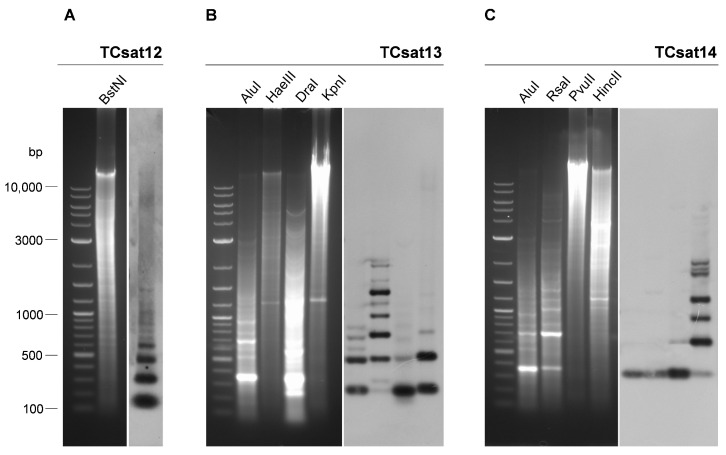
Gel electrophoresis of *T. castaneum* genomic DNA digested with the different restriction endonucleases indicated above the lanes (left panels) and corresponding Southern blots (right panels) obtained after hybridization with probes specific for satDNAs TCsat12 (**A**), TCsat13 (**B**), and TCsat14 (**C**).

**Figure 6 genes-14-00999-f006:**
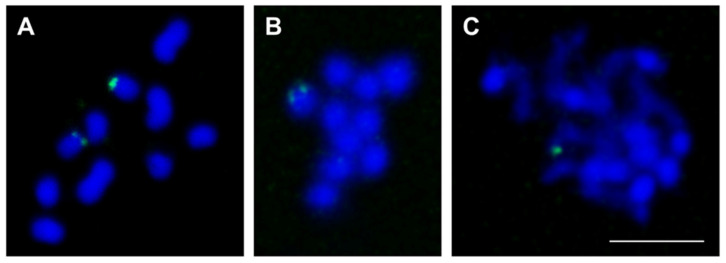
Localization of TCsat12 (**A**), TCsat13 (**B**), and TCsat14 (**C**) satDNAs on *T. castaneum* chromosomes (2n = 20). The position of satDNAs was determined by FISH. Chromosome spreads were stained with 4′,6-diamidino-2-phenylindole (blue fluorescence) and hybridized with satDNA probes visualized by fluorescein isothiocyanate (green fluorescence). The bar represents 5 µm.

**Table 1 genes-14-00999-t001:** The list of *T. castaneum* satDNAs indicating their monomer length, genomic abundance, and presence on chromosomes, unplaced scaffolds, and unplaced singletons. The presence of satDNA in an array of ≥5 tandem repeats is indicated by black squares, whereas white squares indicate scattered repeats (1–4 copies in an array).

SatDNA	Monomer Length (bp)	Genomic Abundance (%)	Chromosome												Original Name ^1,2,3^
			1	2	3	4	5	6	7	8	9	10	Unplaced Scaffolds	Unplaced Singletons	
**TCsat01**	361	17	□	□	□	□		□	□	□	□	□	□	■	TCAST ^1^
**TCsat02**	359	0.38		□	□	□	□	□	□	□	□	□	□	■	TCAST2 ^2^
**TCsat03**	172	1	□	■	■	■	■	■	■	■	■	■	□	■	Cast1 ^3^
**TCsat04**	172	0.5	■	■	■	■	■	■	□	■	■	■	■	■	Cast2 ^3^
**TCsat05**	227	0.2	■	■	■	■	■	■	■	■	■	■	□	□	Cast3 ^3^
**TCsat06**	179	0.5		■	■	□	■	■	■	■	■	□	■	■	Cast4 ^3^
**TCsat07**	334	1	□	■	■	■	■	■	■	■	■	■	■	■	Cast5 ^3^
**TCsat08**	180	0.5		□	■		■	■	□	■			■	■	Cast6 ^3^
**TCsat09**	121	0.2	□	□	■	■	■	□	□	□	□	□	□	□	Cast7 ^3^
**TCsat10**	169	0.2	□	□	■	□	□	■	□	■	■	■	■	■	Cast8 ^3^
**TCsat11**	350	0.2	□	□	■		□	■	□	□	□	■	■	□	Cast9 ^3^
**TCsat12**	154	0.0461						□					■	■	
**TCsat13**	266	0.0184											■	■	
**TCsat14**	309	0.0154			□				■				□		
**TCsat15**	1106	1.6857	□	□	□	■	□	□	□	□	□	□	□	□	
**TCsat16**	330	0.0059											■		
**TCsat17**	120	0.0077			■										
**TCsat18**	73	0.0892	□		□	■	■	□	□		■	□	□	■	
**TCsat19**	618	0.0174			■			■			■		□		
**TCsat20**	144	0.0045		■											
**TCsat21**	102	0.0041			■										
**TCsat22**	281	0.0040			■										
**TCsat23**	164	0.0078			■				■		■		□	■	
**TCsat24**	721	0.0898								□			□	■	
**TCsat25**	258	0.0027			■										
**TCsat26**	361	0.0030			■								■		
**TCsat27**	302	0.0040										■	■		
**TCsat28**	412	0.0036			■										
**TCsat29**	174	0.0028						□				■	■		
**TCsat30**	90	0.0019							■						
**TCsat31**	144	0.0018											■		
**TCsat32**	325	0.0033			■										
**TCsat33**	150	0.0017			■										
**TCsat34**	203	0.0020			■										
**TCsat35**	172	0.0018									■				
**TCsat36**	178	0.0020					□			■					
**TCsat37**	230	0.0470	□		□	□	■	□	■	■	□	□	□	■	
**TCsat38**	178	0.0070					■	□	□			□	■	□	
**TCsat39**	188	0.0018			■										
**TCsat40**	276	0.0020								■					
**TCsat41**	122	0.0016									■				
**TCsat42**	179	0.0016								□	■				
**TCsat43**	178	0.0016			■										
**TCsat44**	311	0.0021				■									
**TCsat45**	260	0.0017			■										
**TCsat46**	142	0.0017						■							
**TCsat47**	282	0.0019				■									
**TCsat48**	156	0.0017							■						
**TCsat49**	354	0.0017								■					
**TCsat50**	168	0.0024			■										
**TCsat51**	353	0.0043						■							
**TCsat52**	184	0.0013										■	■		
**TCsat53**	152	0.0017			■										
**TCsat54**	322	0.0022											■	□	
**TCsat55**	311	0.0023						■							
**TCsat56**	306	0.0024								■					
**TCsat57**	212	0.0013			■										

^1^ [28], ^2^ [30], ^3^ [9].

## Data Availability

The raw *T. castaneum* whole-genome sequencing data were deposited in the Sequence Read Archive (SRA) database under the BioProject study accession number PRJNA606031. The consensus sequences of the 46 novel satDNAs were deposited in the NCBI GenBank database under accession numbers OQ551416–OQ551461.

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
