# Peer review of "The Low-Copy-Number Satellite DNAs of the Model Beetle Tribolium castaneum"

_genes, 2023, doi:10.3390/genes14050999_

Round 1
Reviewer 1 Report
In this study, the authors carry out a nice and detailed study of the satellitome of the beetle T. casteneum using NGS data. They identified 46 new satDNA families, expanding the number of characterized satDNA in this species to 57. They also analyze in silico the chromosome distribution of these satDNA on the chromosomes of this species.
I found that the manuscript is worthy of publication. It is very well written and can be easily followed in spite of the many analyses performed by the authors.
The article could be published in its current form. I only have one suggestion. Would it not have been better to number the new satellites (TCsat12-TCsat57) by their abundance? With this, also table 2 would be easier to follow.
Reviewer 2 Report
The manuscript referenced genes-2341632 and titled “The low-copy-number satellite DNAs of the model beetle 2 Tribolium castaneum” by Tena Gržan and colleagues presents a repertoire of 46 “novel” satellite DNAs, thus updating the knowledge on the satellitome repertoire of the Coleoptera red flour beetle, a model organism for ethological and food safety research. These data were obtained through low coverage genome sequencing and in silico data analysis. Using this approach, the authors found 46 novel low-copy satDNA sequences occupying 2.1% (at least, in my opinion…) of the genome and interesting potential satDNA “seeds”. These works are extremely important to shed light on these important and yet enigmatic sequences of the genomes, being in this case, the only approach available at this moment (as long-read sequencing is not yet available for a routine use) allowing to “see” these low-copy elements, otherwise “hidden” by the other techniques’ resolution. Additionally, the authors analyzed the surrounding landscape and experimentally tested some sat DNAs regarding their nature/organization and physical chromosome location, what I consider extremely important and complementary to the in silico analysis. Finally, I found the discussion on the possible origin of some satellites, highly associated with transposable elements dynamics appropriate and plausible.
For the reasons I pointed out, I think this manuscript is suitable for publication in Genes Journal.
There are however some minor points the authors should address, namely:
- the eventual and most probable existence of polymorphisms regarding the amount and satDNA sequences (specially at the nucleotide level) between individuals from this species. Although I know that GA2 (Georgia 2) strain is highly used in these studies, this issue must be addressed and discussed, as the sequences found as well as their genome representativeness may vary between populations.
- authors should improve the title legend of Figure S7 by referring what the readers are looking at… Physical localization of the TCsat14 satDNA in chromosomes of…..
- I also think Fig. 7 is accessory as it doesn’t add value to the MS.
